# SERVQUAL Method as an “Old New” Tool for Improving the Quality of Medical Services: A Literature Review

**DOI:** 10.3390/ijerph182010758

**Published:** 2021-10-13

**Authors:** Aleksandra Jonkisz, Piotr Karniej, Dorota Krasowska

**Affiliations:** 1Department of Dermatology, Venerology and Pediatric Dermatology, Medical University of Lublin, 20-080 Lublin, Poland; dor.krasowska@gmail.com; 2Faculty of Health Sciences, Wroclaw Medical University, 50-367 Wrocław, Poland; piotr.karniej@umed.wroc.pl

**Keywords:** service quality, SERVQUAL, management at public healthy, service quality improvement

## Abstract

The second half of the 20th century saw the development of a new trend in the management of medical services across Europe. Those shifts were associated with the transformation of various spheres of human life, both on professional and private levels. The service market then turned back to “quality”, already known in antiquity. According to Aristotle, “quality” is one of the basic categories of thought and reality of the human population. The research material was obtained from literature databases, including Scopus, Cochrane, Medline, and PubMed, as well as from literature reports, including monographs, research papers (e.g., doctoral dissertations), and others. The available literature was assessed with regards to the abovementioned objectives of our study and considering possible advantages from its implementation. Therefore, the applied research method was based on a bibliographic query and desktop data analysis. The adopted research methodology hinged on exploration, compilation, analysis, and processing of data and information from available sources, resulting in drawing up of summary conclusions. The obtained data were subjected to reciprocal confrontation with an attempt to evaluate new possibilities of using the method in other medical specialties. The Servqual method enables us to learn the patient’s expectations, while the service provider can identify irregularities and implement corrections. It allows the executive staff of medical facilities to change elements of medical procedures, which improves the quality of the service provided and thus increases the satisfaction and compliance of patients.

## 1. Introduction

The second half of the 20th century saw the development of a new trend in the management of medical services across Europe. Those changes were associated with the transformation of various spheres of human life, both on professional and private levels. The service market then turned back to “quality”, already known in antiquity. According to Aristotle, “quality” is one of the basic categories of thought and reality of the human population.

Currently, according to the International Organization for Standardization (ISO), “quality” is “the totality of properties of a product or service that determine its to meet identified or anticipated needs.” “Quality” is a comparison between expectation and performance or the obtained effect [1]. According to Opolski K. et al., quality is an objective goal that should be pursued [2,3].

“Quality” consists of all the elements of a product, including services that contribute to the satisfaction of a potential customer. Human health care requires the highest quality at every level of services provided. Taking into account the WHO’s definitions of medical service quality, Opolski et al. claim that the quality of medical services provided should be determined by the highest professional competence and dedication that meet the patient’s expectations [2,3]. The quality of medical services can be evaluated in the following two areas: clinical (postulated) quality and perceived quality. The clinical quality of services reflects an objective medical outcome. In turn, the perceived quality is related to the patient’s subjective awareness about the manner in which he/she was contacted, cared for, or shown interest at a medical facility [2,3]. According to Opolski and the team, an important component of the aforementioned quality, i.e., perceived quality, is the patient’s perception of staff competence level, as well as of the convenience and aesthetics of the medical institution [2]. Human health and life quality levels, resulting from high standards of medical services offered and provided on the market, are further important aspects of the quality of medical services. 

Another aspect of the quality of medical services is also extremely important, namely, the level of human health and life as an effect of the quality of the service provided on the medical market. The level of human health and life is the final result of the quality of the service provided on the medical market. The Servqual method fits in with the quality of service management models. There are many models of service quality management, i.e., a series of more or less tangible activities taking place during the interaction between the client and the service provider that are delivered as a solution to the client’s problems. Among them, the concepts of Gronroos, Gummesson, and Berry should be mentioned. They all assume that service quality is the result of the quality expected and obtained. The Servqual (SQ) model, an acronym from the words “service” and “quality”, is based on these premises. 

### 1.1. Servqual: Description and Meaning

SQ, designed to assess service quality and based on standardized parameters, was developed by Zeithamlai, Parasuraman, and Barry for the non-medical sector in 1985 [4].

The authors of the method assumed the existence of gaps (discrepancies) between the levels of service provided and service expected. When a customer’s expectations exceed his/her own real experiences a difference in service quality emerges. An identification of such discrepancies/gaps may help eliminate the dissonance between the level of a customer’s expectations and patient perceptions of service provided, which, in turn, may contribute to increased customer satisfaction and thus improve the quality of service [5]. The quality of the provided medical procedures has priority over other elements of the service (material components of the service, e.g., the technical condition of the building) and is treated as a priority. Quality takes precedence over other elements of service provided and is thus given a priority. The SQ gap model enables the identification of five gaps/discrepancies and the factors that relate to them, making it possible to determine the service quality provided. SQ is therefore intended to assess the level of customer satisfaction with service quality in various sectors or industries [6].

According to Parasuraman A et al., five gaps are distinguished.

The first gap relates to the differences between customer expectations and the perceptions of a generating entity (service providers) towards the needs of customers (e.g., patients). The size of the gap is influenced by marketing research, carried out by a given entity.

The second gap relates to the contradiction between the concept of the service and its factual characteristics. The size of this gap depends on the management’s commitment to service quality issues and setting goals, as well as the standardization of tasks and the perception of opportunities.

The next, the third, gap relates to discrepancies between the service provided and the specificity concerning the creation of the quality of services. The size of this gap depends on teamwork, as well as on the matching of employees to entrusted work, technology, and the perception of control, as well as the supervision and control system.

The fourth inconsistency is the difference between promised and delivered service. The size of this gap is influenced by horizontal communication as well as by tendencies to inflate promises.

The last, the fifth, gap results from the previous gaps, being the difference between what the client (e.g., patient) expects and what he/she receives [4]. The sizes of the above-mentioned gaps are influenced by various factors, including the commitment of manager and employees, marketing research, standardization of activities, the perception of customer requirements, and contacts with the customer. Hence, the comparison of expectations and the perception of service quality give an answer to the question of how service quality is perceived by the customer [6]. 

The first figure shows a simplified diagram of the SQ method (Figure 1).

The SERVQUAL model, which is a research tool, determines the relative impact of five dimensions, namely, tangibility, reliability, responsibility, confidence, and empathy, on customer perception [4]. In the medical field, according to Szyc et al., an efficient identification of errors in the process of creating and providing services is conducive to high quality [7]. The SQ method basically refers to gap 5, the last one in the above list, combining service quality design from the customer’s point of view, where the customer is also the service provider. Service evaluation is carried out by means of a questionnaire/survey, dedicated to this method and regarded as a measurement tool.

The SERVQUAL scale consists of 44 questions, aimed to support the evaluation of the gap between expectations and perceptions. The first 22 questions address the customer’s expectations and the second set of 22 questions enquire about the customer’s perceptions of service provided [5]. The answers to questions are presented in a five-level format of the Likert scale, where 1 is definitely dissatisfied and 5 is definitely satisfied.

The service quality rating is then determined by calculating the difference between the ratings of customer’s perceptions and expectations, according to the formula below:SQ = P − E
where SQ is overall service quality, P is perception of service quality provided, and E is expected service quality. 

A positive assessment of the gap indicates that the client’s expectations have been met, i.e., the perception of services is very high. If, on the other hand, the score for the gap is negative, it means that the services provided have not met the expectations, so their perception is unsatisfactory.

Considering the above, to monitor the quality of medical services provided, it would seem reasonable to use one of the service quality assessment methods used in the medical sector, such as the Servqual model.

### 1.2. The Aim of the Study

The primary objectives of the study included a review of available literature reports, substantiating the Servqual method’s eligibility in assessing the quality of medical services provided by various medical facilities (Hospital ward, Emergency Department, Primary health care, Diagnostic laboratory) and an analysis of how far it could be implemented in daily clinical practice of health care providers. A secondary objective was to define the application pathways of this method in particular medical specialties. The further goal was to determine the possibility of using this method in the daily practice of the listed medical entities.

## 2. Materials and Methods

The study material was acquired from literature databases, including Scopus, Cochrane, Medline, and PubMed, as well as from literature reports, including monographs, research papers (e.g., doctoral dissertations), and others. The available literature was assessed with regards to the abovementioned objectives of our study and considering possible benefits from its implementation.

Therefore, the applied research method was based on a bibliographic query and desktop data analysis. The implemented research methodology hinged on exploration, compilation, analysis, and processing of data and information from available sources, followed by drawing up summary conclusions. The achieved data were subjected to mutual confrontation with an attempt to evaluate new possibilities of using the method in other medical specialties.

The Servqual model is used to analyze the levels of satisfaction with the quality of the services offered by many countries around the world, including European countries. When it is used on the level of medical services, it seeks to determine the level of dissonance, i.e., the gap between what the patient expects and what a health care unit delivers. Such a study offers a substantive evaluation of results and may help execute, for example, appropriate corrective measures to improve the quality of medical care offered. SQ has been utilized equally for surgical and conservative treatment specialties, both in outpatient and inpatient settings.

## 3. Discussions

Research on the quality of medical services, provided by inpatient health care included, inter alia, internal medicine, surgery, gynecology, and hospital emergency departments. The surveyed patients emphasized the competence and kindness of hospital staff, their ability to inspire trust and self-confidence, and their professionalism [5,7,8,9,10]. In 2012, the results of studies, conducted at a surgical department, were published [7]. The analysis covered five areas, characteristic of health care and such elements as qualifications and remuneration of personnel, hospital equipment, and the costs of patient hospitalization [7]. The surveyed patients indicated that the factors, related to costs, especially related to the premises, as well as empathy of the staff, their competences, and communication with patients, had a very significant impact on the size of the studied gap, i.e., gap 5. Finally, it was found that materiality-related factors had had a significant impact on patient satisfaction, from medical service provided. Regarding the cited group of patients, the hospital premises played an important role, followed by factors related to empathy, staff competences, and communication with the patient [7]. A 2016 report by Nadi et al. from a study conducted, among others, at a surgical hospital, showed that the highest priority was empathy, followed by materiality [10].

According to Szyc et al., meeting of the patient’s needs, considering the functional quality, increases their comfort and, consequently, translates into satisfaction with hospitalization [7]. The meta-analysis of Rezaei et al. showed that patient expectations of the health care level, as provided by Iranian hospitals, were not met, while the expectations of those patients exceeded their assumptions of the quality of services provided by the facility [11]. The highest and lowest values of the gap in the qualitative score were related to the dimensions of reliability and responsiveness, respectively [11].

Research, using the gap model and conducted in 2016 among emergency department patients, showed that the most important element, influencing patient satisfaction, was the material factor. That group of respondents considered empathy to be the least important in assessing medical service quality, while assurance, reliability, and responsiveness were more important [8]. Additionally, the analysis of the causes of dissatisfaction of emergency department patients, published in 2015 by Rahymati et al., indicated that about 25% of patients unambiguously assessed the staff’s actions in providing medical services [12].

Questionnaire results, published in 2016 in “Medical Archives”, allowed the Nadi team to identify the priorities that determined the quality of hospital medical services for studied patients, with empathy being on top [10]. In that ranking list, the second, the third, and the fourth priorities were assigned to materiality, responsiveness, and assurance level, respectively, while reliability turned out to be the least important factor [10]. 

Some authors analyzed the factors influencing the assessment of medical service quality by hospitalized patients at private institutions and the public sector. Already at the end of the 1990s, service quality indicators were identified, showing that private hospitals were then expected to provide higher quality services, especially in the domain of hotel conditions [13]. Nevertheless, it was the public sector that far exceeded patients’ expectations [12]. The SQ model was also used as an empirical tool to analyze the quality of services provided by private hospitals in Malaysia in 2003. The results, based on an evaluation of the mean differences between expectations and perceptions, indicated that the perceived value of services by patients exceeded their expectations for all the measured variables [14].

Tolga and Jiju published in 2006 the results of an SQ study in patients of the abovementioned types of medical facilities, which indicated that the patients, hospitalized at private hospitals, were more satisfied with the quality of services than those at public hospitals [15]. In addition, patients at private institutions were also more satisfied with care provided to them by the medical personnel (doctors, nurses) and it was the satisfaction with the work of the medical staff that was the most important determinant of service quality provided [15].

The results, obtained in 2018 by Javed and Ilyas, showed that patient satisfaction with medical service was most strongly associated with empathy in the public sector and responsiveness in the private sector [16]. Hence, according to the researchers, satisfaction with the work of hospital staff and reasonable costs were the most important determinants of service quality at public inpatient care [14,16].

In 2013, Al-Borie et al. published the results of a survey of nearly 1000 patients’ perceptions of the quality of services provided at hospitals (private and public) [17]. The authors pointed out that the material status of the patient as well as his/her occupation had a statistically significant effect on the level of satisfaction with the quality of service. By contrast, the patient’s age was not important in service quality assessment provided. According to the Saudi researchers, the SQ model is a reliable instrument in medical marketing [17].

Some researchers also confirm that the differences in reception of service provided depend on the respondent’s sex, age, and education level [1,17,18]. According to Papanikolaou and Zygiaris, older respondents tend to perceive service quality as higher vs. younger respondents, especially regarding the domains of responsiveness, confidence, and empathy. This is most likely because older people may have had more contacts with the public health sector and, thus, more experience from those periods [1]. On the other hand, young patients are more demanding towards the health care provider. It would be interesting, according to these researchers, to explore how previous experiences and their frequency influence the perceived quality of medical services. Patients with higher education in the cited study expected higher standards of quality level from the provider. Moreover, educated people may seem better prepared to assess the quality of services, while uneducated people may have lower expectations about the quality of medical services provided by medical staff. In the research of Fraihi et al. from 2016, the expectations of women were higher in terms of reliability and materiality than those of men [18]. In addition, there was a significant relationship between age groups and the dimensions of reliability, sensitivity, and empathy: Patients > 73 years demonstrated higher expectations in all the dimensions, while a significant difference was observed in the dimensions of reliability, responsiveness, and empathy when compared to other age groups [18].

Measuring the quality of services at medical facilities is an essential condition for the development of health care systems. Therefore, the analysis of individual dimensions of service quality and the discrepancy between patients’ perceptions and expectations enable a selection of an appropriate remedial strategy for service providers. Raposo et al. emphasized that the perceived quality, both technical and functional, contributed to the improvement of patient satisfaction [19]. Research, carried out in 2020 at a public facility, which was the University Clinical Hospital, resulted in valuable insights on the scope of necessary improvements by monitoring and interpreting the level of identified gaps [20].

Perera and Dabney, analyzing five dimensions of the quality of medical services, using the SQ model, divided them into “tangible” and “intangible”. The “intangible dimensions” included reliability, responsiveness, confidence, and empathy. According to the abovementioned authors, the “intangible” dimensions exert a significant impact on both the perception of quality and patient satisfaction with offered medical services [21]. The size of the gap in reliability has a significant impact on both the overall quality of the service and patient satisfaction, while discrepancies in empathy have a significant impact on satisfaction but not on the overall quality of the service provided [21].

Interesting observations were presented by Ramirez, together with Pineda, on the perception of medical service quality provided, viewed through the framework of the health care provider’s infrastructure [22]. Such a tangible/material gap discourages patients from using it. In turn, Zarei et al.’s research project from 2015 showed that the quality of tangible/material factors did not have any significant impact on the assessment of service quality provided, thus showing a different opinion in their conclusions [23].

A nuclear medicine center was another type of medical facility, i.e., outpatient, from which an analysis of medical service quality was published [24,25]. It turned out that patients considered materiality and certainty as one dimension, while the primary dimension of empathy was divided into empathy and comfort [24]. According to De Man et al., the perception of the quality of services by patients correlated with patient satisfaction, especially in terms of reliability and substantive confidence of the service provider [24]. That enabled them to define a strategy for improving the operation of a medical facility, thus maximizing the professional competences and technical skills of the staff. Observations of outpatients showed that the total subjective waiting time for a service had a greater impact on the reliability dimension than on other dimensions of service quality provided in the Servqual model. Providing information by medical personnel, e.g., about the reasons for delays of medical procedures, had a significant impact on the perception of the healthcare provider’s reliability by patients [25]. A need to verify the quality of provided medical services resulted in a reassessment of patient satisfaction with the services of a Pakistani rural health center after a 1-year period of changes introduced by the healthcare provider. Based on the SQ questionnaire, gaps were identified in the areas of empathy and reliability. At that time, 1533 patients were surveyed, and the obtained observations and the resulting changes provided a gradual increase in patient satisfaction with medical services at that facility [26].

The SQ instrument was also used to assess the level of patient satisfaction with obstetric services [9]. The results, published in 2014, showed that women’s satisfaction with the medical services provided to them increased with the age of respondents, the number of children they already had, and the number of visits. It was also confirmed in that case that the lower was the educational status, the higher was the assessment of the quality of medical services provided [9]. The most powerful predictor of satisfaction of the surveyed women with service quality provided was communication with the service provider, i.e., medical staff, and the speed of their actions. Therefore, according to Ali M et al., targeted training of service-providing staff, sensitizing them to the needs of patients, contributes to improved quality of offered medical services [9].

In a cross-sectional study by Frsihi et al. in Saud Arabia, regarding outpatient care, the results showed that patients’ expectations exceeded their perception in all the dimensions of service quality, pointing to statistically significant gaps [18]. Additionally, a meta-analysis, published by Teshnizi et al., showed that patient perception expressed dissatisfaction in all the dimensions of service quality assessed by the SERVQUAL tool. Out of the five dimensions, responsibility and reliability showed the largest gaps [27].

Babakus and Mangold suggested in their 1992 study that low ratings of perceived quality by patients signaled the existence of deeper problems underlying the service provided [28]. Thus, they may reflect the inability of a medical facility to hire and retain high-quality specialists or even to provide adequate training for staff. The World Health Organization (WHO) emphasized the importance of quality in provided health care, in accordance with the criteria of efficiency, cost, and social acceptance for good standards of medical services and of the staff they are provided by.

The SQ method, perceived as a tool for examining the quality of medical services, has been used for many years and the results it produces enable problems to be identified and changes to be made, which is consistent with the suggestions of many organizations acting for the benefit of patients’ well-being, e.g., WHO. The Servqual model has practical applications as a management tool and enables characterizing the organizational factors responsible for the failure to meet expectations of service users. In Bebko’s opinion, the quality of service can be measured by the level of discrepancy between the expectations or desires of recipients and their perception of what they have received, whether there is a gap model [29]. The research results, presented in the literature, show that it is an effective and stable tool, intended to measure the quality of services in various sectors, including the medical one [16,18,30].

The Servqual method is also a useful instrument to assess the quality of medical services for their improvement in longer perspectives. It is used to identify quality factors and measure patient satisfaction of various medical service-providing entities, both in- and outpatient. Moreover, it can be successfully used to monitor the quality of medical services. According to Papanikolaou and Zygiaris, it enables estimating the level of satisfaction but also allows defining the dimension in which experience exceeds expectations and the dimension in which experience does not meet expectations [1]. It is believed that the quality of healthcare should be redefined, considering its multidimensionality. In addition to the usual discrepancy in the assessment of service provided, patients may possess and express different understandings of healthcare. This has a significant impact on their perceptions of the quality of medical services. The analysis of the patient’s satisfaction level by the healthcare provider is a difficult task because each patient perceives the quality differently and has different expectations towards medical services offered.

### Limitations in Servqual Method

Some researchers point to some limitations of the Servqual method and emphasize that it does not show consistent results in terms of the internal content of the scales [4,31,32,33,34]. At the root of these discrepancies there are, among others, cultural differences in the way people understand and give meaning to the social world and medical care [2,3]. Babakus and Boller indicate that SQ is characterized by a poor fit, and the obtained results do not meet the more stringent convergence criteria and contribute to the discrimination of specific variances [35]. In their opinion, SQ can only be used to develop a one-dimensional assessment of the quality of services and not their multi-dimensional assessment. These authors also point to necessary caution when interpreting obtained results by means of different scales, as the results may be distorted. Furthermore, the formulation and definition of the context of the results, obtained by the Servqual method, may also be a problem [35]. A critical look at the Servqual method is also presented by Van Dyke et al. [32]. They indicated that the difficulties, related to the application of the method, could be divided into four categories. The first category includes problems with using differences or vulnerability points. The second category is the allegation of the model credibility, and the third one is an ambiguous definition of the concept of “expectation”. The last, i.e., the fourth, category addresses the unstable dimensionality of the Servqual instrument.

Cwiklicki believes that this model does not work, among others, for the services linked to a product [31]. The author emphasizes that the quality of services, provided in individual dimensions, can be grouped according to the criteria of the degree of meeting customer expectations. The scale (high, medium and low) corresponds to the size of the gap [31]. Other reservations concern, inter alia, the use of the same questionnaire for different types of services, the discrepancy between expectation and perception cannot be obtained by measuring only the perception of clients, or blurring the difference between the subjectivity of feeling and the mastery of execution [34,36].

Interesting enough are the insights of Mauri et al., who analyzed nearly 30 years of research of a gap model in international academic databases. However, despite some critical theoretical–conceptual and methodological–operational aspects, the gap model and the SERVQUAL scale are still the most frequently used instruments for service quality studies, met in marketing literature [37]. The medical service market is highly specific, being, on the one hand, the subject of market regulations (supply/stay), and of the market rules of competition for patients, on the other. The basic feature of medical service is its immateriality, as shown by the SQ model, together with the identification of several implications for the management of medical facilities.

## 4. Conclusions and Results

The mentioned above considerations are conceptual work, the aim of which was to probe/learn/determine the use of the SQ method in the identification of factors influencing the assessment of the quality of services by patients. The Servqual model has been successfully used over many years to assess the quality of medical services in countries with different levels of economic development. Although it has both advantages and disadvantages, it is also distinguished by contemplating customer expectations, categorization of research and outcomes. SQ is widely used by various industries and sectors. Its application allows for identifying quality gaps and targeting restorative changes. A strategy leading to the improvement of medical services’ quality, regardless of the type of applied examination, contributes to the health improvement, limitation of complications, and faster returns of recovering patients to professional activities. In the literature on this subject, one can meet, however, with a criticism of SQ. Despite the critical opinions, Servqual is an essential tool to assess the quality of medical services, as well as an important source of information about patients’ expectations. Medical services should be of high quality, which, in turn, should be properly defined. The Servqual method facilitates the provider to learn the patient’s expectations and can identify irregularities to implement corrections. The use of the SQ method gives great possibilities of increasing the quality of the service offered, but at the same time has shortcomings. The shortcomings include the use of the same questionnaire pattern for various types of services, as in the case of medical services, e.g., it is confirmed in hospital and outpatient treatment. Likewise, the concept of the SQ method does not take into account the distorting difference between the bias of emotion and the proficiency of the performance of a given service and, therefore, examines only some of the attributes of a given service. The service quality management maximizes the professional skills of medical personnel, improving the timeliness of services provided and improving the flow of information between medical personnel and the patient, which improves the professionalism of the service provided. This also relates to the fact that all repair procedures should be re-verified at specified intervals, i.e., reminder training for service providers, e.g., every 2 years. An important value of the work added that can be utilized in the hospital practice is the use of quality improvement of the tool that already exists for the efficient organization and functioning of medical services. Hospitals are looking for solutions proven in various sections of the economy, which, when adjusted to the medical environment, will allow enhancing the quality of medical services offered. The SQ method is definitely a method that, in spite of the fact that it is based on the assumptions of the economic environment, after being implemented, it effectively improves the quality of medical services. In the context of guidelines for future research in the application of the method, the authors suggest that it is reasonable to search for individual applications for each medical specialty. The variety of the patient’s needs, as well as the various ways of organizing the medical service, determines the need to adapt the method to specific services each time. The SQ method, despite its reputation, requires suitable implementation.

## Figures and Tables

**Figure 1 ijerph-18-10758-f001:**
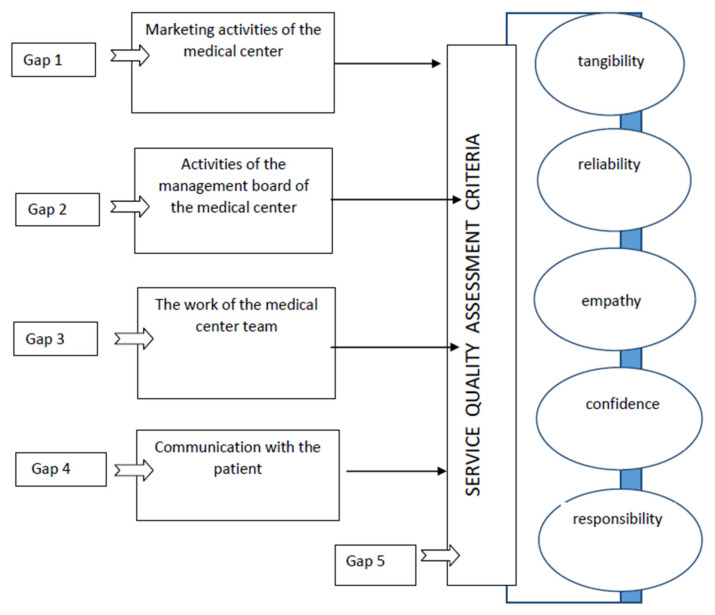
Quality gap model, on the basis of Parasuraman A. et al. “A Conceptual Model of Service Quality and Its Implications for Future Research” [4], used with the permission of the publisher of the original article.

## Data Availability

Not applicable.

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
