# Peer review of "SERVQUAL Method as an “Old New” Tool for Improving the Quality of Medical Services: A Literature Review"

_ijerph, 2021, doi:10.3390/ijerph182010758_

Round 1
Reviewer 1 Report
This study had provided a literature review of SERVQUAL for improving the quality of medical services. Some major suggestions are that the authors need to give more details:
The article lacks a clearly defined aim of the research. It is not clear the authors aimed to compare the different ways of using SERVQUAL or to report the outcomes of SERVQUAL in medical services. The authors had not provided details of SERVQUAL applications among the literature and it not easy to see the improvements of SERVQUAL in measuring medical services as time goes on. The authors had also not compared the applications of SERVQUAL in different medical services.
The discussion looks like a report not like a real discussion. It would be expected to see that which limitation of SERVQUAL tool was found in measuring medical services and how the later literature improved the SERVQUAL tool. The authors did point some limitations of SERVQUAL tool but had not focus on medical services and not provide more discussion about the improvements.
The conclusions are too weak. What is the contribution of the article? What are policy implications for medical centers? How the results contribute to the practice? What are the limitations of this study? What are the suggestions for the future research?
Author Response
Dear Revwier 1,
Thank you very much for your comments. Because of them I could improve my paper. Please find attached the answer.
Best regards,
Dr Aleksandra Jonkisz

Reviewer 2 Report
The manuscript establishes a literature review for the quality of medical services. The research material was acquired from literature databases, including Scopus, Cochrane, Medline, PubMed, etc. Overall, this is an interesting line of research that is neat and important.
The authors are advised to consider the following suggestions to enhance the study further.
1)The content of the abstract needs to be polished. Please add some appropriate management implications to the abstract.
2)In section 1.2 states about this study purpose(objective), the relevance of the Servqual method’s eligibility in assessing the quality of medical services is not clear. Sufficient elaboration on this section would be beneficial.
3) In spite, the authors present five gaps to explore the relevant literature, but why does the “smart healthcare” or “smart medical” not include the research gaps scope. Because intelligent health is an important issue that needs to be considered.
4)It is recommended the authors summarize the topics of the presented research dimension in the table.
5) The “conclusion” is rather too sketchy; please combine the section of management implications and more details about the main results. Adequate discussions maybe have practical value.
Author Response
Dear Reviewer 2,
Thank you very much for your correct comments. Please find attached the answer. I hope that the table has met your expectations.
Kind regards,
Dr Aleksandra Jonkisz

Reviewer 3 Report
Overall, an interesting review. However, further detail is required, particularly in the methods and results section and the discussion would benefit from being reframed. There are some minor issues with language and use of terms that are inappropriate for good academic writing. Some specific comments provided below.
Introduction: It would read better to say “service provided” rather than “provided service” throughout the introduction. E.g. “discrepancies between the service provided and…”
Lines 35-36: Punctuation missing.
Line 47: I dont think these are the only factors in patient satisfaction - what about the way they are treated, the care and kindness of staff, availability of services, accessibility...?
Line 51: What is "service quality management"? Does this refer to health services? Or other services? This is a broad term and needs to be further defined in context.
Line 60: What do you mean by exceeds their “imagination”? I think it should read that their expectations exceed the actual experiences.
Paragraph 1.1: It would be beneficial to avoid using “his/her” and rephrase your sentences. For example: in Line 61 you could say “…and perceptions of…” rather than having the his/her included. It reads more professionally.
Line 64: What does quality take precedence over? This sentence is vague and I don’t see how it applies to health care. Perhaps “quality is the underpinning value of health service delivery”.
Lines 78 and 85: Please refrain for saying “i.e.” – this is implied.
Figure 1 is not referenced or discussed in your text. The title of this figure is not well phrased. Please include this to show the reader what the figure is for and how it fits with your review.
Line 97: Are you talking specifically about health/medical services? Or all services? This section is not well focused.
Line 107: Should be “is” not was as you are describing how it is performed – not how you performed it.
Methods: Was this a systematic review or other type of literature review? Did you use a PRISMA? What criteria were used for your search? Key words? How many people performed the search and the analysis? A lot more detail Is needed in this section.
Where are your results? How many articles were sourced and used? How were articles excluded?
Discussion: Without a results section, it is difficult to follow the discussion. However, you have included some valid findings but provided limited discussion around your findings. I found it to be descriptive rather than analytical, particularly Lines 147 - 220. More discussion about the meaning of your findings would make the paper more valuable. It might be beneficial to divide your discussion up into themes or sections such as: Internal factors (intrinsic expectations), External factors (facility). Also, refraining from using subjective terms such as “interesting enough” will improve your discussion.
Author Response
Dear Reviewer 3,
Thanks a lot for your helpful and valuable comments. I hope my answers met your expectations.
Kind regards,
Dr Aleksandra Jonkisz

Round 2
Reviewer 1 Report
No more comments.
Author Response
Dear Reviewer,
Thank you very much for '' no more comments''. Thank you for your work and all the tips given. I saw you marked "Extensive editing of English language and style required" so my colleague helped me especially in the conclusions section. I am sending a new version. I hope now is more readable.
Dr. Aleksandra Jonkisz

Reviewer 3 Report
This paper has been improved, however there are still some minor language changes to be made. Also, the methods section is not appropriate for a literature review. As per my previous comments, this section should include a detailed description of how your literature review as performed and what type of literature review was utilised (e.g. scoping, narrative etc...). There should be a results section that shows how many articles were selected for inclusion and why, as well as those that were excluded and why. This is where your key terms would be beneficial as well your inclusion criteria and exclusion criteria.
Also, see lines 76. 78. 81 and 92. The wording should be changed to have "provided" second (not first) as per my previous comments.
Figure 1 title is not appropriate. This figure should be discussed in the results section - this is where you highlight your own interpretation and explain why you have created this figure - as you indicate that it is as a result of your interpretation of the literature you reviewed.
Line 391 - what does "repair possibilities" mean? Repair is not the appropriate term. This might be a language issue as I note it is used again. I would suggest having someone review all the language used throughout the article as there are many instances where wording is not correct.
Please take your time to go through your article and make changes to improve the feasibility of your study and the readability. Use the methods section to describe exactly what you did and devise a results section to highlight what you found in terms of articles.
Author Response
Dear Reviewer,
Thank you very much for all valuable and inspiring comments. Hope you find the attached table with answers to your questions comprehensive and satisfactory.
Kind regards,
Dr. Aleksandra Jonkisz
